# Autoencoders and Generative Adversarial Networks for Imbalanced Sequence Classification

## Abstract

We introduce a novel synthetic oversampling method for variable length, multi-feature sequence datasets based on autoencoders and generative adversarial networks. We show that this method improves classification accuracy for highly imbalanced sequence classification tasks. We show that this method outperforms standard oversampling techniques that use techniques such as SMOTE and autoencoders. We also use generative adversarial networks on the majority class as an outlier detection method for novelty detection, with limited classification improvement. We show that the use of generative adversarial network based synthetic data improves classification model performance on a variety of sequence data sets.

Dealing with imbalanced datasets is the crux of many real world classification problems. These problems deal with complex multivariate data such as variable length, multi-feature sequence data. Canonical examples can be found in the finance world, for example, questions related to stock market data of several securities or credit card fraud detection often deal with sequence data with many features. Other imbalanced data problems include questions in the medical field such as tumor detection and post surgery prognosis (Zięba et al., 2014). In each of these problems, false positives are more desirable than false negatives, they require sequential data, and the classes are imbalanced.

Class imbalances in datasets oftentimes lead to increased difficulty in classification problems as many machine learning algorithms assume that the dataset is balanced. There are two general approaches to improve classification accuracy for unbalanced datasets. One method is algorithmic, for example, a modified loss function can be used so that misclassifications of minority labeled data are penalized more heavily than misclassifications of majority labeled data (Geng & Luo, 2019). The other is to decrease data imbalances in the training set either by ensembling the data or by generating synthetic training data to augment the amount of data in the minority set.

This motivates the development of methods to improve classification accuracy on variable length, multi-feature sequence data. Given a sequence of $T$ feature vectors, we want to predict labels of the sequence. Oftentimes it is not obvious how to apply methods for unbalanced data to sequence data in a way that takes advantage of the fact that sequential events have the potential to be highly correlated. SMOTE (Chawla et al., 2002) is widely used for oversampling, but does not capture the sequential dimension. Enhanced Structure Preserved Oversampling (ESPO) (Cao et al., 2013) allows one to generate synthetic data that preserves the sequence structure, however it requires that the feature vector has only a single feature at each of the $T$ time points and that the output label is a scalar. As there is no obvious extension to the case where there are multiple features at each time point and the output is also a sequence of labels, the situations where ESPO can be applied are limited.

We develop a method based on deep learning models for sequences in order to decrease data imbalances of sequence data with an arbitrary number of features. We call each feature vector, $x_i \in R^n$, an event in the sequence. We consider the use of generative adversarial networks (GANs) to generate synthetic data. Here, we build a generative model that generates both the feature vectors in a sequence as well as the corresponding labels. We benchmark this synthetic data generation technique against a number of models. We demonstrate that the model trained on the GAN based synthetic data outperforms the baseline model, other standard synthetic data generation techniques, and a GAN based novelty detection method. For each of the synthetic data generation methods, we

train a sequence-to-sequence model (Sutskever et al., 2014) on the dataset that outputs a sequence with the same length as the label sequence. In addition to benchmarking against existing synthetic data generation techniques, we also train a model on the unaugmented dataset. All of the models are embedded within the standard ensemble approach. On all of our datasets, we observe that the GAN based synthetic data generation model significantly improves over the baseline models by 15% to 127% depending on the dataset, while the novelty detection based GAN performs similarly to the baseline model.

The main contributions are as follows:

1. a novel synthetic data generation technique that uses a GAN with an Autoencoder component to generate synthetic data for variable length, multi-feature sequential data in a way that preserves the structure of sequences for both feature vectors and labels;

2. a new novelty detection method for sequential data that uses a GAN as an outlier detection function;

3. a computational study of existing imbalanced classification techniques on highly imbalanced sequential datasets.

In the next section, we discuss relevant literature. Section 3 discusses all of the models, while the computational results are presented in Section 4.

# 1 LITERATURE REVIEW

Many methods exist for imbalanced data. The majority of these methods are developed for non-sequential data and generally take one of two approaches. The first approach is algorithmic and either involves altering the loss function or performance metric in a way that emphasizes the correct classification of the minority set. The second approach is to decrease the data imbalance either by resampling or by generating synthetic minority data such that the training data is more balanced.

The benefit of using algorithmic methods is that they have a straightforward application to sequence data as we can calculate the loss and accuracy the same way for both a vector and a scalar. Methods that are commonly used include a weighted loss function in which the loss of misclassifying minority data is greater than the loss of misclassifying majority data (Sun et al., 2007; Geng & Luo, 2019). We implement a weighted loss function in all our models.

In contrast to the algorithmic methods, we can instead consider data level methods that strive to balance the two classes. There have been many different methods that are developed to balance the dataset without generating synthetic minority data. Since these methods alter how the training set is built, applying them to sequence data is straightforward. Both ensembling and data sampling techniques fall under this category. Ensemble methods take the original training set and build subsets of the training set such that the sizes of the minority and majority sets are more balanced (Galar et al., 2012). On the other hand, other methods for dataset creation involve over- or under-sampling (Kubat & Matwin, 1997). Ensemble methods generally outperform over- and under-sampling methods alone so we use ensembles in all our experiments.

Another data level method that can mitigate the class imbalance problem is to generate synthetic minority data. SMOTE (Chawla et al., 2002) is one of the most widely used methods for generating synthetic minority data. For this method, synthetic data is generated via interpolation between nearest neighbors in the minority set. There are many extensions to SMOTE that aim to increase classification performance by sharpening the boundary between the two classes. One such example is ADASYN (He et al., 2008), which explores the composition of the nearest neighbors to determine how many synthetic data points to generate and how to generate them. Neither SMOTE nor ADASYN cannot be used to oversample sequence data because these methods build a synthetic feature vector by independently interpolating between the real data points, so the framework cannot capture correlation in time. However, methods have been developed that use an autoencoder and apply SMOTE in the latent space in order to oversample sequence data (Lim et al., 2018).

Structure Preserving Oversampling and ESPO are the only methods, to the best of our knowledge, that exist for dealing with unbalanced sequence data (Cao et al., 2011; 2013). To generate synthetic sequence data, these methods use the covariance structure of the minority data in order to build

synthetic minority data that captures the sequential structure. They are developed for single feature sequences and there is not a straightforward extension to data that has multiple features for each event. This is because we cannot calculate the covariance matrix for each feature independently since features may interact with each other in different ways at different events.

Another method for synthetic data generation are GANs (Goodfellow et al., 2014). This model pits a generator model, which generates synthetic data, and a discriminator model, which tries to distinguish between real and synthetic data, against each other. By pitting the models against each other, it trains both the generator and discriminator, and once the generator has been trained, we can use it to generate synthetic minority data. While this approach has been applied to oversample both image data (Zenati et al., 2018; Guo et al., 2019; Douzas & Bacao, 2018) and sequence data (Yu et al., 2017), they have not yet been developed to oversample both sequence data and labels. GAN based models designed for sequence data have been used for synthetic text generation, but as this architecture is not designed for classification, the sequence class is not considered. These models cannot generate both a sequence and the associated labels. GAN based models have been used to build imbalanced sequence classification models, but the benefit of generating GAN-based synthetic minority data is that it allows for flexibility during classification model selection. (Rezaei et al., 2018).

Both SMOTE and GAN based synthetic data generation techniques have been shown to improve classification performance for certain types of highly imbalanced datasets such as image data or single feature sequences. These models have not yet been developed to sequence data with an arbitrary number of features as even methods developed for generating sequential synthetic data cannot deal with sequence data with more than one feature. GAN based models cannot be directly applied to synthetic minority data generation as the output from the generator is an embedding of the input sequence. So while these methods improve a classifier's performance, unlike the other data-level methods and the algorithmic methods, they have not yet been developed and applied to generic sequence data.

Historically, anomaly detection methods generally use a model such as PCA or SVM to determine which data points are outliers and thus are more likely to be in the minority class (Schölkopf et al., 2000; Hoffmann, 2007; Ma & Perkins, 2003; Shyu et al., 2003). However, novelty detection methods can be improved by the use of more complex outlier detection methods. In deep learning, various LSTM based autoencoder models have been used in novelty detection methods for sequence data so that the outlier detection model can exploit the structure of the data (Marchi et al., 2015; 2017; Principi et al., 2017; Schreyer et al., 2017). For the same reason, GANs have also been used for novelty detection methods for both image and sequence data (Wang et al., 2018; Chang et al., 2019a; Rajendran et al., 2018; Chang et al., 2019b).

## 2 APPROACHES

We assume that we have sequences $x = (x_1, \ldots x_T) \in \mathcal{X}$ and associated labels $y = (y_1, \ldots, y_L) \in \mathcal{Y}$ where each $x_i$ has $n$ features and $L$ labels to predict. Each of the labels $y_\ell$ for $\ell \in L$ is a class label, either 0 or 1. We consider binary labels at each prediction step, but multi-class labels can be considered as well. Sequence length $T$ can vary by sequence. We also assume there is a dominant label sequence called majority and all other label sequences are minority. Since we focus on minority sequences, all our synthetic oversampling methods also work with no modification in the presence of multiple majority classes. For the baseline model, we consider a sequence-to-sequence (seq2seq) architecture. This is an encoder-decoder architecture where the entire sequence is represented by an $s$ dimensional hidden vector $h_T^0$, the encoder hidden state at the final event. We then use this vector, $h_T^0$, as the input to the decoder model at each event. The model can be written as

$$h_t^0 = f_{\theta_E}^0(h_{t-1}^0, x_t), \ t \in [1, T]$$
$$h_\ell^1 = f_{\theta_D}^1(h_{\ell-1}^1, h_T^0), \ \ell \in [1, L]$$
$$o_\ell = \text{softmax}(h_\ell^1)$$

where $f_{\theta_E}^0, f_{\theta_D}^1$ are cell functions such as LSTM or GRU and $o_\ell$ is the $\ell^{th}$ predicted label (Sutskever et al., 2014). In our experiments, we use a seq2seq model with attention (Bahdanau et al., 2014) and a weighted loss function where the weights are proportional to class balance as the classification

method. The output of this seq2seq model is of the same length as the label sequence. We ensemble the data into $K$ ensembles where each ensemble contains a subset of the majority data and all of the minority training data and in inference, we average the predictions from each ensemble. In order to evaluate the synthetic data generation techniques, we train seq2seq models both with and without synthetic minority data and compare the results.

## 2.1 ADASYN ON AUTOENCODERS

In a straightforward application of SMOTE to sequences, we reshape $x$ to a vector and then apply the SMOTE algorithm directly to $x$. In addition, by reshaping the label $y$, we can interpolate between the label vectors associated with the samples used to generate the synthetic sample. This creates a fractional valued label that has to be converted to a binary one if the underlying model requires it. However, this method can only be applied to sequences of the same length since it does not make sense to interpolate between variable length inputs. We compare the straightforward SMOTE application on the datasets where sequences are all of the same length. In order to provide a baseline to compare the GAN based synthetic minority technique against, we consider how SMOTE can be applied to variable length sequences. We discuss the how ADASYN can be applied to variable length sequences and its advantages over SMOTE in Appendix A.1.

## 2.2 GENERATIVE ADVERSARIAL NETWORK BASED TECHNIQUES

### 2.2.1 GAN BASED SYNTHETIC DATA

We develop a GAN that is capable of generating both sequences, $x$, and associated label vectors $y$. As in any GAN model, we must build both a generator and a discriminator and train the models by pitting them against each other. The model that we discuss is based on the improved Wasserstein GAN (Gulrajani et al., 2017; Arjovsky et al., 2017). Recall that in the standard baseline classification model, we use a seq2seq model to get sequences $h_x$ and $h_y$ of hidden states from sequences $x$ and labels $y$, respectively. For the generator model, $G_{\phi_{EN_1}, \phi_{EN_2}}(z, x, y)$ we use a seq2seq model with LSTM cells to get hidden state sequences $h_x$ and $h_y$. We include an addit ional argument $z$ to initialize the cell state for the generator. For the true data, we set $z$ to 0 and for the fake data we use $z \sim \mathcal{N}(0, I)$. The model is able to distinguish between $x$ and $y$ since $x$ is the input for the generator encoder and $y$ is the input for the generator decoder. The parameters $\phi_{EN_1}$ and $\phi_{EN_2}$ correspond to $x$ and $y$, respectively. The discriminator model, $D_{\phi_{D_1}, \phi_{D_2}}(h_x, h_y)$ uses a seq2seq model trained on the hidden sequences $h_x$ and $h_y$ to get a real valued output, $c$. As in the generator, $\phi_{D_1}$ are parameters corresponding to $x$ and $\phi_{D_2}$ to $y$. The loss function compares the outputs from the discriminator model for the real and fake data.

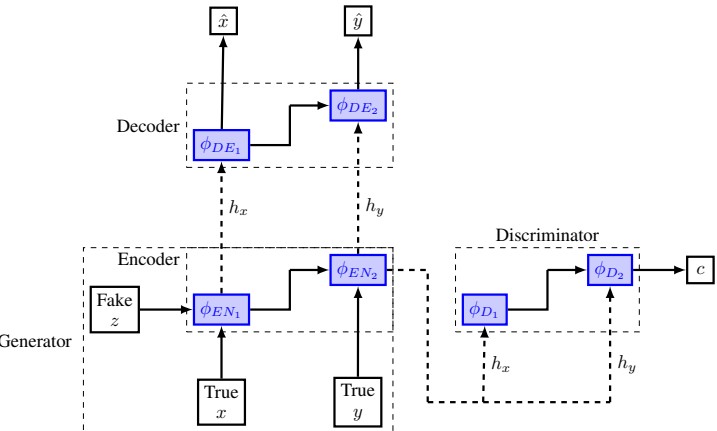

Figure 1: Overview of GAN model. Sequences and labels are used as input to GAN and both the discriminator and decoder use the outputs from the generator model.

We also need a component of the model to decode $h_x$ and $h_y$ in a meaningful way. Therefore, we have a seq2seq based autoencoder, $A_{\phi_{EN_1}, \phi_{EN_2}, \phi_{DE_1}, \phi_{DE_2}}(x, y)$, that takes as input $x$ and $y$,

creates hidden sequences $h_x$ and $h_y$, and then reconstructs $\hat{x}$ and $\hat{y}$. The autoencoder shares the encoding part with the generator. This GAN architecture differs from existing GAN-based synthetic data generation methods as each of the three components of the GAN with Autoencoder model are comprised of LSTM encoder-decoder architectures in order to generate both minority sequences and associated labels.

In Figure 1, the GAN with autoencoder structure is sketched out. For model training, we use the loss function

$$
\begin{aligned}
\mathcal{L} &= \mathbb{E}[D_{\phi_{D_1}, \phi_{D_2}}(G_{\phi_{EN_1}, \phi_{EN_2}}(z, x, y))] - \mathbb{E}[D_{\phi_{D_1}, \phi_{D_2}}(G_{\phi_{EN_1} \phi_{EN_2}}(0, x, y))] \\
&+ \lambda \mathbb{E}\left[\left(\left\|\nabla D_{\phi_{D_1}, \phi_{D_2}}(G_{\phi_{EN_1} \phi_{EN_2}}(0, x, y))\right\|_2 - 1\right)^2\right] \\
&+ \mu \mathbb{E}\left[\left\|(x, y) - A_{\phi_{EN_1}, \phi_{EN_2}, \phi_{DE_1}, \phi_{DE_2}}(x, y)\right\|_2^2\right]
\end{aligned}
\tag{1}
$$

where $\lambda$ and $\mu$ are tunable hyperparameters. All expectations are with respect to the minority sequences $(x, y)$.

During training, we want to prevent the discriminator from learning too quickly so that the generator can learn. We use Adam (Kingma & Adam, 2015), and set the discriminator learning rate lower than the generator learning rate to prevent the discriminator from learning too quickly. To further slow down discriminator training, we add noise to generator outputs and decrease the noise as model training progresses (Chintala et al., 2016). We monitor generator, discriminator and autoencoder loss during training and adjust $\mu$ and $\lambda$ to prevent the discriminator from learning too quickly and to ensure that the autoencoder loss decreases during training.

During model training, we train the generator, discriminator and autoencoder weights on different batches of data. We first update the weights associated with the generator, $\phi_{EN_1}$ and $\phi_{EN_2}$, by considering all terms in the loss function. Next, we update the weights associated with the discriminator, $\phi_{D_1}$ and $\phi_{D_2}$, by including the first three terms of the loss function as the autoencoder loss term does not depend on the discriminator weights. Finally, we update the weights associated with the decoder part of the autoencoder, $\phi_{DE_1}$ and $\phi_{DE_2}$, using the last term of the loss function. The weights of the encoder part of the autoencoder are shared with the generator, so they are not updated along with the rest of the autoencoder weights. For datasets with a single label prediction, we consider a GAN with autoencoder model, where instead of a seq2seq architecture for each of the model components, we use LSTM cells and the input to the generator is $x$ and $z$. We then assign the minority label to generated minority data.

Once we have trained the generator in conjunction with the discriminator and autoencoder, we can use the generator and the decoder part of the autoencoder to generate synthetic minority data. As this model is trained only on the minority dataset, we require a reasonably sized minority training set. In our experiments, we consider minority training sets with at least 1000 samples. We generate 3 synthetic samples from each minority sample in the training dataset by feeding in vectors $z \sim \mathcal{N}(0, 1)$ into model and using the autoencoder output as synthetic minority data. We expect that with random noise $z$ will slightly perturb the minority data in order to generate novel synthetic minority samples instead of simply oversampling existing minority data. This method should improve on the ADASYN with autoencoder model as it allows for the simultaneous generation of both the sequences and associated label vectors. We discuss how this model can be used to for novelty detection when trained on the majority data in Appendix A.2.

## 3 COMPUTATIONAL STUDY

We consider three imbalanced datasets[1]. Each of these datasets consists of multi-feature sequence data where the data imbalance is less than 5% (it can be as low as 0.1%). The first dataset is a proprietary medical device dataset where the data comes from medical device outputs. The second dataset we consider is a sentiment analysis dataset that classifies IMDB movie reviews as positive

---

[1]Code and data are available at `to-be-added`

or negative (Maas et al., 2011). Though the data is initially balanced, for this paper, we downsample the positive class in order to use it for an anomaly detection task. Lastly, we consider a power consumption dataset[2] where the goal is to predict if voltage consumption changes significantly. A class corresponds to whether the voltage change is considered significant. For the medical device dataset and IMDB sentiment dataset, we make a single label prediction and thus we consider the seq2one model for both these datasets. For the power consumption dataset, we consider both the seq2seq and seq2one tasks to show that the GAN with autoencoder generated synthetic data improves model performance in both cases. For each dataset, we report the minority class F1 score on the test set. If there are multiple minority classes, we report the average F1 score of the minority classes. Details of model implementation are available in Appendix B and additional performance metrics are available in Appendix B.1.

## 3.1 MEDICAL DEVICE DATA

In this dataset, the data is a sequence of readouts from medical devices and the labels indicate if a user error occurs. The sequence length is on average 50 and there are around 50 features. We have on order of 1 million samples and less than 1% of the samples are from the minority class. We make 5 runs, each one with a different seed, and thus each run has different ensemble models.

Table 1: Test F1-Scores for Each Seed

| Run | Baseline | GAN-based Synthetic Data | ADASYN Autoencoder | GAN Discriminator Novelty Detection | GAN Autoencoder Novelty Detection |
|---|---|---|---|---|---|
| 0 | 0.79% | **2.02%** | 0.52% | 0.50% | 1.27% |
| 1 | 1.77% | **3.15%** | 0.30% | 0.50% | 1.14% |
| 2 | 1.28% | **2.06%** | 0.50% | 0.32% | 1.26% |
| 3 | 1.29% | **1.72%** | 0.49% | 0.50% | 1.00% |
| 4 | 0.68% | **1.79%** | 0.52% | 0.50% | 1.17% |
| Average | 1.16% | **2.15%** | 0.47% | 0.46% | 1.17% |
| Standard Deviation | 0.44% | **0.58%** | 0.09% | 0.08% | 0.11% |

Comparing the results of each of the proposed methods against the baseline in Table 1, we observe that the only method that significantly improves classification accuracy is the GAN-based synthetic data model with p-value = 0.01 based on the t-test. Surprisingly, using the ADASYN Autoencoder generated synthetic data leads to a substantial decrease in the F1-score, suggesting that this synthetic data technique does not capture the structure of the minority data. This suggests that interpolation in the autoencoder latent space is not sufficient, and the GAN component of the autoencoder is necessary. We also note that the difference in the F1-score between the two novelty detection methods is significant with p-value=2.8e-6 according to the t-test. We observe that the choice of outlier detection is important for novelty detection.

Table 2: Differences Between Predictions for GAN Minority and Baseline Models

|  | True Majority | True Minority |
|---|---|---|
| Predicted Majority | 60 | -1 |
| Predicted Minority | -60 | 1 |

To explore how the models trained on the synthetic data improve on the baseline models, we examine the difference between the confusion matrix of predictions on the test set for a model trained with and without the GAN-based synthetic data. In Table 2, we note that a number of false negatives and false positives in the baseline model are converted to true positives and true negatives, respectively in the model trained on the GAN-based synthetic data. That is, the improvement in classification accuracy of the model trained with the GAN-based synthetic data is due to a decrease in both false negatives and false positives.

Examining the classification of true minority and synthetic minority samples in the GAN-based synthetic data training set, we observe that the trained model is better at correctly classifying the

---

[2]https://archive.ics.uci.edu/ml/datasets/individual+household+electric+power+consumption

synthetic minority samples than the true minority samples which is interesting. For run 0, the F1-score for the true minority training samples is 0.4036 while the F1-score for the synthetic minority training samples is 1. This also reveals that the model overfits since the test F1-score is much lower. This is not surprising for such a heavily imbalanced dataset.

## 3.2 SENTIMENT

We consider all reviews under 600 words long and front pad reviews so that all samples in our dataset are of length 600. We then use the GoogleNews trained word2vec model to embed the dataset. In order to make this dataset imbalanced, we downsample the positive reviews to create two datasets where the positive reviews comprise 1% and 5% of the training set respectively and then ensemble the training dataset. The resulting dataset is comprised of around 25 thousand samples with 20% in test. Training models on this dataset is computationally expensive because of the sequence length, so we only consider a single run for these experiments.

Table 3: Test F1-Scores

| Data Imbalance | Baseline | GAN-based Synthetic Data | ADASYN Autoencoder | GAN Discriminator Novelty Detection | GAN Autoencoder Novelty Detection |
|---|---|---|---|---|---|
| 1% | 7.80% | **17.76%** | 0.00% | 2.36% | 1.86% |
| 5% | **56.75%** | 52.85% | 9.47% | 9.63% | 9.46% |

In Table 3, we compare the results of each of the proposed methods against the baseline. The only method that significantly improves the F1-score is the model trained on the GAN-based synthetic data. We also note that with 5% imbalance, the baseline model performance on the ensembles is high enough that the anomaly detection methods we consider do not improve performance. This suggests that these synthetic data generation techniques are only effective for highly imbalanced datasets.

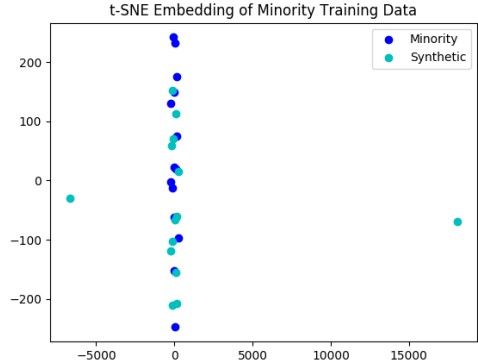

Figure 2: t-SNE Embedding of Minority Data

For this dataset, we conclude that 5% imbalance is an upper bound for which the proposed anomaly detection techniques can be used. However, studying classification of true minority and synthetic minority samples in the GAN-based synthetic data, we notice that the trained model correctly identifies all minority samples in the training set, both true and synthetic. This suggests that the sentiment analysis task is an easier task.

To understand how well the GAN-based synthetic data training set is able to capture the structure of the minority data, we use t-SNE to embed a subset of the true and synthetic minority training data so it can be visualized. In Figure 2, it is clear that the true minority data falls along a line and all but two synthetic minority samples also fall along the same line. As the synthetic samples are staggered along the line, it suggests that for the most part, the synthetic minority data successfully mimics the minority data. Additional t-SNE plots are available in Appendix C.1 as t-SNE embeddings can vary from run to run.

### 3.3 POWER

We use a dataset of power usage in a given household in trying to predict if voltage usage changes significantly. Sequences are of length 20 and there are 6 features. We have around 2 million sample and approximately 2% of the samples are in the minority class. As this dataset is not padded, we compare our GAN-based synthetic data technique against a model trained with SMOTE generated synthetic data.

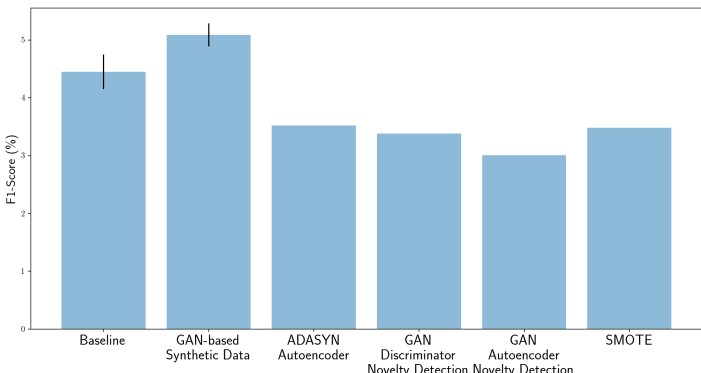

Figure 3: Bar Plot of Test F1-Scores for Each Model with Confidence Intervals for Models Trained on Multiple Datasets

Comparing the results of each of the proposed methods against the baseline in Figure 3, we conclude that the only method that significantly improves the F1-score is the model trained on the GAN-based synthetic data. To test the significance of this improvement, we generate ensembles using 5 different seeds and train a baseline and GAN-based synthetic data model on each run. In the five runs, the average baseline F1-score is 4.51%, the average F1-score for the GAN-based synthetic data is 5.10%, and the improvement with the GAN-based synthetic data is significant with p-value=0.016 based on the t-test.

Note that the relative difference in the F1-score between the baseline model and the GAN-based synthetic data model is about 15% and lower than either the Medical Device or Sentiment dataset. As the Power dataset has fewer features than the other two datasets, we observe that the GAN-based synthetic data is better able to capture the data structure for more complex sequences.

On this dataset, we also consider sequences where the associated label vectors are of length 4 by predicting if the voltage change is significant for 4 time periods. As before, sequences are of length 20. We consider a sample as minority if the voltage change is significant in any of the 4 time periods. Approximately 7% of the data is in the minority class. We only consider the GAN-based synthetic data model on this dataset as it is the only model that improves on the baseline in Figure 3. The average baseline F1-score is 0.25% and the average F1-score for the GAN-based synthetic data is 0.59%. Though the imbalance is lower, it is unsurprising that the F1-score is so low as we are making 4 predictions for each sequence. We do not do multiple runs for this dataset as the relative F1-score increase is high. We conclude that the GAN-based synthetic data can be used to improve model performance for datasets with label sequences.

## 4 CONCLUSIONS

We have presented several techniques for synthetic oversampling in anomaly detection for multi-feature sequence datasets. Models were evaluated on three datasets where it was observed that GAN-based synthetic data generation outperforms all other models on all datasets. We also note that GAN-based synthetic data yielded larger classification F1-score increases over other models for datasets with more features. Furthermore, we provide evidence that the GAN-based synthetic data is capable of capturing the structure of minority data. We also demonstrate that GAN-based synthetic data generation techniques can be applied to datasets with label sequences. Finally, we provide evidence that synthetic oversampling is beneficial for datasets with substantial imbalances (less than 5% in our datasets).

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

## A  APPROACHES

### A.1  ADASYN ON AUTOENCODERS

We discern how to use an autoencoder and ADASYN to generate synthetic data. We first train an autoencoder on minority data. Using the trained autoencoder on the minority data, we obtain $h_T^0 \in \mathbb{R}^s$ for each sequence. Once we have embedded the sequence, we can then run the SMOTE algorithm to get $\hat{h}_T^0$. Next, we can use the decoder half of the autoencoder to lift $\hat{h}_T^0$ back to $\hat{x}$. The benefit of this approach is that the encoded minority data captures the structure of the sequence. All that remains is to generate the associated labels for the synthetic data.

An approach is to find a way to use the weights for interpolating between the minority data in the SMOTE algorithm to generate the associated label vector via interpolation. Based on SMOTE for sequence $x^i$, given $\left(h_T^0\right)^i$ and $\left(h_T^0\right)^j$ obtained from $x^i$ and $x^j$, respectively, a synthetic sample

$$\left(\hat{h}_T^0\right)_{\text{syn}} = \left(h_T^0\right)^i + \mathbf{w}^i \odot \left(\left(h_T^0\right)^i - \left(h_T^0\right)^j\right)$$

is generated where $\left(h_T^0\right)^j$ is one of the neighbors of $\left(h_T^0\right)^i$ and $\mathbf{w}^i = (w_0^i, \ldots, w_s^i)$ are fixed weights with $\odot$ representing component wise multiplication. Note that this equality does not hold for $\hat{x}$, $x^i$, and $x^j$ where $\hat{x}$ is generated by the decoder with respect to $\left(\hat{h}_T^0\right)_{\text{syn}}$. We then generate the associated label vector as

$$\hat{y} = y^i + \bar{w}^i(y^i - y^j).$$

where $\bar{w}^i = \frac{1}{s}\sum_{j=1}^s w_j^i$. The downside to this approach is that if $w_j^i$ is a uniformly chosen random number in [0,1], then $\bar{w}^i \sim 0.5$ for $s$ large. Therefore, instead of considering the SMOTE algorithm in conjunction with the autoencoder, we consider the ADASYN algorithm instead. There are two main differences between ADASYN and SMOTE. Instead of choosing weights $w_j^i \sim U[0,1]$, we choose a single random interpolation weight, $w^i$, for each synthetic sample. In addition, the number of synthetic sequences to generate from each sequence in the minority set is adaptively chosen. The label vector, $\hat{y}$, associated with $\hat{x}$ is defined as

$$\hat{y} = y_i + w^i(y_i - y_j).$$

This method then allows us to apply ADASYN to sequences in a way that should both preserve the structure of the data and generate both sequences and labels.

### A.2  GAN NOVELTY DETECTION AND GAN DISCRIMINATOR DETECTION

While the previous section trains the GAN model on the minority data, in novelty detection, GAN is trained only on the majority data. One approach to novelty detection is to examine the autoencoder reconstruction loss. When computing the autoencoder reconstruction loss on the trained model, we expect the reconstruction loss be higher for the minority class than for the majority class. Similarly, we can examine the discriminator output of the trained model. Unlike existing GAN based anomaly detection methods for sequences (Chang et al., 2019b), this model does not depend on the autoencoder reconstruction loss to train the generator, but instead allows for the use of other discriminator functions. This flexibility allows for the use of different GAN architectures such as improved Wasserstein GAN (Gulrajani et al., 2017).

We expect that the minority class data should be classified as fake data by the discriminator, while the majority class data would be classified as real data. However, since the novelty detection prediction with GAN model on majority data from Figure 1 requires the label vector, $y$, it needs to be modified. The model is similar to the model sketched out in Figure 1, except that we use LSTM cells to get the sequence $h_x$ of hidden states from sequences $x$, and the discriminator and autoencoder take as input $h_x$ and $x$, respectively. The generator takes as input noise $z$ and sequence $x$. The loss function used to train this GAN model is similar to the loss function in (1) and it is trained by using the same logic as the GAN-based synthetic data model. Basically, the model is the same except that the labels $y$ are neglected. We can then use either the autoencoder or the discriminator of this GAN model to classify the majority and minority classes in a novelty detection method. Note that this approach only infers minority/majority classification and not the actual labels $y$.

## B  COMPUTATIONAL STUDY

For each of the datasets, the data is ensembled into 10 ensembles such that each ensemble contains all of the minority data and a random subset of the majority data. Sequences in each dataset are front-padded to the maximum sequence length for model training. The GAN based oversampling and novelty detection methods are implemented using Tensorflow and the remaining models are implemented using Keras with Tensorflow. We use the Adam optimizer for the GAN based models (Chintala et al., 2016; Radford et al., 2015), while for the remaining models, we use the Adadelta optimizer (Zeiler, 2012) in model training. All models are trained on a single GPU card. For each dataset, we tune the number of layers and number of neurons of the baseline model. We use the best performing model as the baseline for comparison.

### B.1  ADDITIONAL PERFORMANCE METRICS

In addition to reporting the F1-score, we also consider the G-mean and PR AUC metrics for both the baseline model and the model trained with GAN-based synthetic data in order to get a complete picture of how the two models compare. We do not consider the additional metrics on the remaining models as they underperform the baseline model.

### B.1.1  MEDICAL DEVICE DATASET

(a) Test G-mean for Each Seed

| Run | Baseline | GAN-based Synthetic Data |
|---|---|---|
| 0 | 17.0% | **24.1%** |
| 1 | 17.7% | **24.2%** |
| 2 | 23.4% | **24.1%** |
| 3 | 24.0% | **24.1%** |
| 4 | 17.0% | **24.1%** |
| Average | 19.8% | **24.1%** |
| Standard Deviation | 3.18% | **0.04%** |

(b) Test PR AUC for Each Seed

| Run | Baseline | GAN-based Synthetic Data |
|---|---|---|
| 0 | 0.0025 | **0.0031** |
| 1 | 0.0030 | **0.0036** |
| 2 | 0.0027 | **0.0031** |
| 3 | 0.0028 | **0.0029** |
| 4 | 0.0025 | **0.0030** |
| Average | 0.0027 | **0.0031** |
| Standard Deviation | 0.0002 | **0.0002** |

We see in Table 4a and Table 4b that the model trained on the GAN-based synthetic data outperforms the baseline model on both the G-mean and PR AUC metrics.

### B.1.2  SENTIMENT DATASET

(a) Test G-mean for Each Seed

| Data Imbalance | Baseline | GAN-based Synthetic Data |
|---|---|---|
| 1% | 22.7% | **56.4%** |

(b) Test PR AUC for Each Seed

| Data Imbalance | Baseline | GAN-based Synthetic Data |
|---|---|---|
| 1% | 0.031 | **0.062** |

We see in Table 5a and Table 5b that the model trained on the GAN-based synthetic data outperforms the baseline model on both the G-mean and PR AUC metrics for the 1% imbalance. We do

not consider the G-mean or PR AUC metrics for the dataset with 5% imbalance as the GAN-based synthetic data does not improve classification accuracy for that level of data imbalance.

### B.1.3 POWER DATASET

| (a) Test G-mean for Each Seed | | |
|---|---|---|
| Run | Baseline | GAN-based Synthetic Data |
| 0 | 17.3% | **23.0%** |
| 1 | 17.3% | **20.0%** |
| 2 | 16.6% | **20.4%** |
| 3 | 16.1% | **19.0%** |
| 4 | 14.2% | **21.9%** |
| Average | 16.3% | **20.9%** |
| Standard Deviation | **1.3%** | 1.6% |

| (b) Test PR AUC for Each Seed | | |
|---|---|---|
| Run | Baseline | GAN-based Synthetic Data |
| 0 | 0.02 | 0.02 |
| 1 | 0.02 | 0.02 |
| 2 | 0.02 | 0.02 |
| 3 | 0.02 | 0.02 |
| 4 | 0.02 | 0.02 |
| Average | 0.02 | 0.02 |
| Standard Deviation | 0 | 0 |

We see in Table 6a and Table 6b that the model trained on the GAN-based synthetic data outperforms the baseline model on the G-mean metric, but not the PR AUC metric. It is interesting that there is a much larger disparity in both the G-mean and F1-scores between the baseline model and the model trained with GAN-based synthetic data, yet the PR AUC scores are identical.

## C RESULTS

### C.1 T-SNE EMBEDDINGS FOR MINORITY SENTIMENT DATA

For the Sentiment dataset, we generate synthetic minority samples and embed both the real and synthetic minority data in 2-dimensional space in order to visualize the data. We run the t-SNE embedding with 5 different random seeds as the embeddings can vary from run to run. We see that in each of the figures below, that the majority of the synthetic minority samples are very similar to the real data for each of the runs. We also note that in all the runs that at most two samples are located away from the main cluster of the real and fake minority samples.

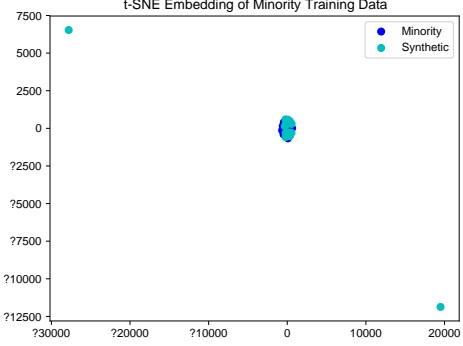

Figure 7: t-SNE Embedding of Minority Data with Random Seed 0

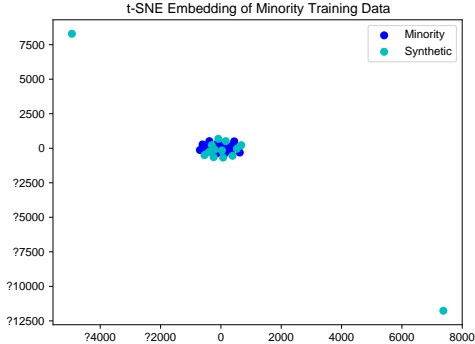

Figure 8: t-SNE Embedding of Minority Data with Random Seed 1

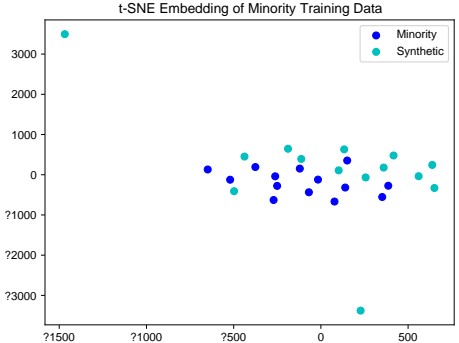

Figure 9: t-SNE Embedding of Minority Data with Random Seed 2

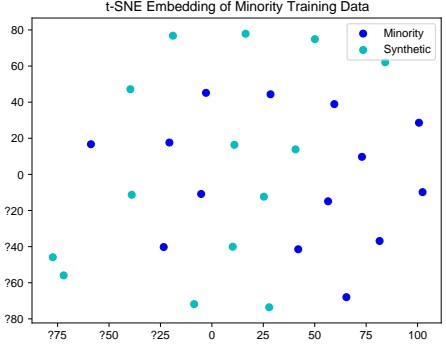

Figure 10: t-SNE Embedding of Minority Data with Random Seed 3

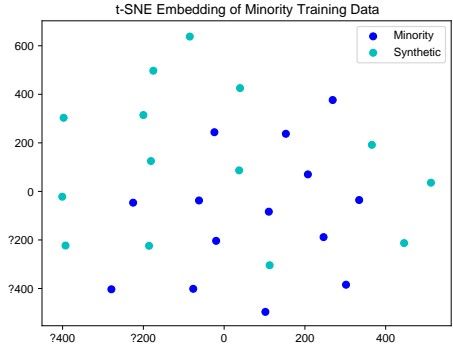

Figure 11: t-SNE Embedding of Minority Data with Random Seed 4

