# OpenReview forum: "Autoencoders and Generative Adversarial Networks for Imbalanced Sequence Classification"
_ICLR.cc/2020/Conference — Reject_

### Official Review · AnonReviewer3 · 2019-10-21
**Official Blind Review #3**

**Rating:** 3

**Review:**

The paper consider important and interesting problem: how to generate a sequence from minority class if we want to do oversampling with synthetic data in a way similar to SMOTE.

Now from the paper, the exact used approach is not clear, as the details are too scarce (see some examples below). Experiments are not convincing, as the authors don't compare to the state of the art approaches. As the exact problem statement is hard to grasp, it is hard to identify the exact contribution of authors.
Note, that the proposed approaches, in my opinion, are not that different from approaches from modern data generation for imbalanced classification [1, 2], as we propose some kind of GAN to generate new data and so have only two variables: how we select loss for this GAN and how we select the architecture.

I assume, that to be accepted at a major venue a deeper investigation is required at the moment.

See also the following comments:
1. Figure 4: title is not required, as we have a caption with the same information. Better to use confidence bars too.
2. The figures will benefit from usage of vector format.
3. Figure 3: tSNE can vary from one run to another. It is better to provide at least 3 random figures or even better train e.g. a simple classification model for t-SNE embedded model and present ROC AUC scores for identification synthetic/non-synthetic.
4. Table 1&2 formatting is different from that usually used in Academy (see e.g. https://dl.sciencesocieties.org/files/publications/style/chapter-05.pdf)
5. F1 score is often not the best metric for imbalanced problems. The paper will benefit from providing also PR AUC (average precision) scores.
6. Figure 2: avoid confusion matrices presented in this form, as they take much space providing almost no information. Classic tables are better.
7. From the problem statement at the very beginning of section 2 it is not clear what kind of labels do we expect (I suppose that for each sequence we have a specific label i.e. all y_i are 0 but one, that is 1)
8. Sometimes bigger weights for minority objects or dropping significant part of majority sequences are enough, so results for these approaches also should be included
9. How the hyperparameters mu and lambda were selected?

[1.] Guo, Ting, et al. "Discriminative Sample Generation for Deep Imbalanced Learning." Proceedings of the 28th International Joint Conference on Artificial Intelligence. AAAI Press, 2019. IJCAI 2019, https://www.ijcai.org/proceedings/2019/0334.pdf
[2.] Douzas, Georgios, and Fernando Bacao. "Effective data generation for imbalanced learning using conditional generative adversarial networks." Expert Systems with applications 91 (2018): 464-471.

**Experience Assessment:**

I have published one or two papers in this area.

**Review Assessment: Checking Correctness Of Derivations And Theory:**

I carefully checked the derivations and theory.

**Review Assessment: Checking Correctness Of Experiments:**

I carefully checked the experiments.

**Review Assessment: Thoroughness In Paper Reading:**

I read the paper at least twice and used my best judgement in assessing the paper.

---

> ### Author Response · Authors · 2019-11-13
> **Clarification for our models**
>
> Please let us know if you have any suggestions for state of the art approaches to compare our methods against. We would appreciate and welcome any suggestions.
>
>  I took a look at both papers you linked, and while I agree that the use of GANs to generate synthetic data is not new, I believe that our model architecture does differ from existing GAN synthetic data generation techniques. Our model allows for the generation of synthetic labels and data for imbalanced sequence-to-sequence tasks, which we believe other synthetic data generation methods do not consider. Thank you for pointing out our lack of clarity about our model architecture, we will make revisions to the paper in order to provide more details.
>
> I believe we were not clear about how we evaluated our proposed methods for synthetic data generation. For each dataset, with or without synthetic data, we trained the same classification model in order to evaluate the impact of the synthetic data. The sequence-to-sequence model we consider as our classification model is implemented with a weighted cost function in order to more heavily penalize minority class misclassification. For each dataset, we ensemble the data and include all the minority data and a subset of the majority data in each ensemble. For the baseline comparison, we consider a model trained without synthetic dataset. Hopefully this helps clarify any confusion you might have about our classification model and how we evaluated our methods.
>
> Lastly, thank you for pointing out our formatting errors and for giving us a ways to improve the clarity of our figures and tables. We will address these concerns in a forthcoming revision.

---

### Official Review · AnonReviewer1 · 2019-11-01
**Official Blind Review #1**

**Rating:** 3

**Review:**

The paper is well-written. The idea is good, but it seems like GANs have been suggested for Imbalanced data sequences before. A quick search on google, I found this paper:
"Multi-Task Generative Adversarial Network for Handling Imbalanced Clinical Data" by Mina Rezaei et al., arXiv:1811.10419v1

Moreover, the paper doesn't seem to be comparing their results with other state of the art imbalanced sequence classification methods. The comparisons are all between different proposed GAN methods.

For these two reasons, I do not recommend this paper for publication at this point.

**Experience Assessment:**

I have read many papers in this area.

**Review Assessment: Checking Correctness Of Derivations And Theory:**

I assessed the sensibility of the derivations and theory.

**Review Assessment: Checking Correctness Of Experiments:**

I assessed the sensibility of the experiments.

**Review Assessment: Thoroughness In Paper Reading:**

I read the paper at least twice and used my best judgement in assessing the paper.

---

> ### Author Response · Authors · 2019-11-12
> **Request for Clarification/Suggestions for Baseline Comparison**
>
> Thank you for your comments. Do you have suggestions for imbalanced multivariate sequence classification methods that we could benchmark our methods against? We would appreciate and welcome any suggestions.
>
> I took a look at the paper you suggested and while the authors do use a GAN for imbalanced time series classification, the methods differ because we do not use the GAN model as the classification model, but as a method for generating synthetic data. This then allows us to use a variety of different classification models (in our case a sequence-to-sequence LSTM model) with the augmented dataset. It is true that the idea of using GAN based models to generate synthetic data is not new, but previously it has not been used for multivariate sequence data.

---

### Official Review · AnonReviewer5 · 2019-11-04
**Official Blind Review #5**

**Rating:** 3

**Review:**

1. For imbalanced learning problem, Precision cannot play a good role. Therefore, I recommend using the performance metrics F-value and G-mean to provide comprehensive assessments.
2. In table 2 of the 5%  data imbalance, the proposed method is not as good as the baseline. Could you provide more results with different percentage of data imbalance, such as 2%, 3%, and 4%.
2. The authors created their own baseline, and compared against it. There is plenty of baseline methods in literature to compare against such as:
[1] Nitesh V Chawla, Kevin W Bowyer, Lawrence O Hall, and W Philip Kegelmeyer. SMOTE: synthetic minority over-sampling technique. Journal of Artificial Intelligence Research, 16:321–357, 2002.
[2] Haibo He, Yang Bai, Edwardo A Garcia, and Shutao Li. ADASYN: Adaptive synthetic sampling approach for imbalanced learning. In 2008 IEEE International Joint Conference on Neural Networks, pp. 1322–1328. IEEE, 2008.
[3] Han H, Wang W Y, Mao B H. Borderline-SMOTE: a new over-sampling method in imbalanced data sets learning[C]//International conference on intelligent computing. Springer, Berlin, Heidelberg, 2005: 878-887.
3. Figure 2 is hard to understand.
4. What if projecting both original data and synthetic data into 2D space for visualization, as shown in “Model-Based Oversampling for Imbalanced Sequence Classification”.
5. How robust is the proposed algorithm when facing different levels of noise?


**Experience Assessment:**

I do not know much about this area.

**Review Assessment: Checking Correctness Of Derivations And Theory:**

I assessed the sensibility of the derivations and theory.

**Review Assessment: Checking Correctness Of Experiments:**

I carefully checked the experiments.

**Review Assessment: Thoroughness In Paper Reading:**

I read the paper thoroughly.

---

> ### Author Response · Authors · 2019-11-13
> **Request for Clarification**
>
> Currently, we only provide the F1-score as the performance metric for our models, but you raise a good point about reporting other metrics such as PR AUC or G-mean in order to get a comprehensive understanding of model performance.
>
> In addition, we agree that it would make sense to consider different levels of data imbalance to see how our proposed model performs at different levels of data imbalance. However, training the GAN model to generate synthetic data is computationally expensive and requires some experimentation with hyperparameters, so we will unfortunately not be able to provide additional results.
>
> As the model we are proposing is a method for generating synthetic minority training data, as a first pass, it seems reasonable to compare the model trained with the augmented dataset against a model without any synthetic data. You raise a good point about using SMOTE and related methods directly on the data. We consider a method where we apply ADASYN to the Autoencoder embedding, but we have not considered applying these methods directly to the flattened input. We will run a model trained on an dataset augmented SMOTE generated synthetic data and revise the paper.
>
> While the “Model-Based Oversampling for Imbalanced Sequence Classification” paper projects data into 2D space for visualization using PCA, we project the synthetic and original data for only the IMDB dataset into 2D space using TSNE. We believe that these two approaches are reasonably comparable, but agree that as TSNE runs can vary from run to run, that it makes sense to elaborate more on the 2D visualization and include figures from multiple runs.
>
> Lastly, could you please clarify your comment about different levels of noise? Are you talking about the noisy initial hidden state for the GAN based synthetic data generation?

---

### Decision · Program_Chairs · 2019-12-19

**Decision:**

Reject

**Comment:**

This paper presents a synthetic oversampling method for sequence-to-sequence classification problems based on autoencoders and generative adversarial networks.

All reviews reject the paper for two main reasons:
1 The novelty of the paper is not enough for ICLR as the idea of utilizing GAN for data sampling is common now.
2 The experimental is not convincing as authors did not compare with other leading oversampling methods.

The rebuttal did not well answer these two questions; thus I choose to reject the paper.